# The Potential Mechanisms behind Loperamide-Induced Cardiac Arrhythmias Associated with Human Abuse and Extreme Overdose

**DOI:** 10.3390/biom13091355

**Published:** 2023-09-06

**Authors:** Hua Rong Lu, Bruce P. Damiano, Mohamed Kreir, Jutta Rohrbacher, Henk van der Linde, Tamerlan Saidov, Ard Teisman, David J. Gallacher

**Affiliations:** Global Safety Pharmacology, Janssen Research and Development, Turnhoutseweg 30, 2340 Beerse, Belgium; bpdamiano@anchorpharm.net (B.P.D.); jrohrbac@its.jnj.com (J.R.); hvdlinde@its.jnj.com (H.v.d.L.); t.a.saidov@gmail.com (T.S.); ateisman@its.jnj.com (A.T.); dgallach@its.jnj.com (D.J.G.)

**Keywords:** sodium current, HERG current, ventricular tachycardia (VT) torsade de pointes TdP, loperamide overdose/abuse, conductions slowing, free therapeutic plasma concentration (FTPC), safety margin

## Abstract

Loperamide has been a safe and effective treatment for diarrhea for many years. However, many cases of cardiotoxicity with intentional abuse of loperamide ingestion have recently been reported. We evaluated loperamide in in vitro and in vivo cardiac safety models to understand the mechanisms for this cardiotoxicity. Loperamide slowed conduction (QRS-duration) starting at 0.3 µM [~1200-fold (×) its human Free Therapeutic Plasma Concentration; FTPC] and reduced the QT-interval and caused cardiac arrhythmias starting at 3 µM (~12,000× FTPC) in an isolated rabbit ventricular-wedge model. Loperamide also slowed conduction and elicited Type II/III A-V block in anesthetized guinea pigs at overdose exposures of 879× and 3802× FTPC. In ion-channel studies, loperamide inhibited hERG (I_Kr_)_,_ I_Na,_ and I_Ca_ currents with IC_50_ values of 0.390 µM, 0.526 µM, and 4.091 µM, respectively (i.e., >1560× FTPC). Additionally, in silico trials in human ventricular action potential models based on these IC50s confirmed that loperamide has large safety margins at therapeutic exposures (≤600× FTPC) and confirmed repolarization abnormalities in the case of extreme doses of loperamide. The studies confirmed the large safety margin for the therapeutic use of loperamide but revealed that at the extreme exposure levels observed in human overdose, loperamide can cause a combination of conduction slowing and alterations in repolarization time, resulting in cardiac proarrhythmia. Loperamide’s inhibition of the I_Na_ channel and hERG-mediated IKr are the most likely basis for this cardiac electrophysiological toxicity at overdose exposures. The cardiac toxic effects of loperamide at the overdoses could be aggravated by co-medication with other drug(s) causing ion channel inhibition.

## 1. Introduction

Loperamide is a commonly used and well-established treatment for the symptomatic control of acute and chronic diarrhea and has been available initially as a prescription drug (1976) and subsequently as an over-the-counter medication (since 1988) for over 35 years. Its mechanism of action is based on µ-opioid receptor activation in the gastrointestinal tract [1], which results in antisecretory activity beginning at low therapeutic doses. This µ-opioid agonism is accompanied by the inhibition of the release of acetylcholine and prostaglandins with the consequent reduction in peristalsis and increased intestinal transit time at higher therapeutic doses [2,3]. Local gut opioid receptor binding limits systemic exposure and pronounced first-pass metabolism. Loperamide is also a substrate for P-glycoprotein transporter (P-gp)-mediated efflux, preventing entry to the brain [4]. Even at high clinical doses, loperamide is devoid of central opiate effects, although in rare cases, effects on the central nervous system (CNS) after loperamide intake in children under 3 years old exist [5,6]. Therefore, throughout its extensive 47-year history of use, loperamide has shown an excellent safety profile when used at recommended doses [2,7,8].

However, over the past years, there have been increasing reports of the intentional ingestion of large overdoses of loperamide by opiate addicts that are substantially above recommended therapeutic doses [9,10,11,12,13,14]. Case studies reported some loperamide users took >100 pills/day for 1–4 years before succumbing to cardiotoxicity [15,16]. According to a paper in the Journal of American Pharmacists Association [17], 21 cases were reported between 1985 and 2013 and 33 cases were reported from 2014 to 2016. There has been a large increase in the number of cases of cardiac issues with loperamide abuse over the period between 2015 and 2020 [14]. An unexpected consequence of the abuse of loperamide is a risk of severe cardiac toxicity consisting of pronounced electrophysiological abnormalities and life-threatening arrhythmias [12,18,19,20,21,22,23,24,25,26,27,28,29,30,31]. Common attributes of this cardiac toxicity include the prolongation of both the QTc interval (which includes QRS within its measurement) and QRS duration, AV block/Right Bundle Branch Block, with incidences of torsade de pointes (TdP), including Brugada-like syndrome and non-TdP-like forms of ventricular tachycardia (VT) [31,32,33,34,35]. Other potential confounding risk factors were identified in many cases, e.g., the concurrent use of other drugs with known cardiac actions, such as sotalol, amitriptyline, fluoxetine, clonazepam, methadone, and alprazolam, as well as the use of other drugs of abuse, or the use of low serum potassium [18,23,24,36,37], which may also contribute to the cardiac events that occur alongside excessive loperamide ingestion.

Although other in vitro studies have attempted to identify cellular mechanisms for these potential proarrhythmic actions of loperamide at high doses, the exact mechanisms of cardiac electrophysiological toxicity and arrhythmia with loperamide abuse and extreme overdose are not fully understood. Other studies included an investigation of the effects of loperamide and its active metabolite on in vitro human heterologously expressed cardiac ion channels [38,39,40,41], showing that loperamide inhibits I_Kr_ (hERG: the human *Ether-à-go-go*-Related Gene) with reported IC_50_ values of 33, 40, 54, and 88 nM, and inhibits cardiac voltage-gated sodium channel (I_Na_) with IC_50_ values of 2900 nM (HTS IonWorks in CHO cells) and 239 nM (manual patch clamp in HEK293 cells). In general, drug-induced hERG inhibition has been associated with QT prolongation and the risk of torsade de pointes [42,43] and I_Na_ inhibition with QRS widening, AV blockade, and ventricular tachycardia and fibrillation [38,44,45,46].

For this publication, we used all available internal preclinical data from in vitro and in vivo models to investigate the potential mechanisms for loperamide cardiotoxicity at extreme over-dosing and its safety margin in humans. Specifically, we evaluated the concentration-dependent effects of loperamide on human cardiac ion-channel currents, I_Kr_ (hERG), I_CaL_ (voltage-gated calcium channel-Cav1.2), and I_Na_ (Nav1.5), a sensitive isolated rabbit ventricular wedge preparation and the dose-exposure-dependent effects of loperamide in an anesthetized guinea pig model. Additionally, we used in silico modeling in human ventricular cell populations to predict cardiac safety margins and cardiac toxicities associated with abuse and extreme over-dose, based on the available ion-channel information. In addition, we also used an in silico model to investigate whether the safety margin of loperamide was significantly altered when combined with another treatment affecting ion-channel function (i.e., hydroxyzine, as an example of the combined therapy with loperamide).

## 2. Materials and Methods

All experiments involving the use of animals have been conducted in accordance with the European directive of 2010 (2010/63/EU) on the protection of animals used for scientific purposes and the Belgian Royal Decree of 29th May 2013. Furthermore, these studies were conducted in an AAALAC accredited animal facility.

### 2.1. Compounds

Loperamide was supplied by Janssen Pharmaceutica N.V., (CAS number: 53179-11-6 (made in Beerse, Belgium for the hERG study, or purchased from Sigma-Aldrich, St. Louis, MO, USA), and verapamil (CAS number: 52-53-9) and dofetilide (CAS number: 115256-11-6) were obtained from Sigma (Sigma-Aldrich, St. Louis, MO, USA). Compounds were dissolved in dimethyl sulfoxide (DMSO) (Merck-Sigma Aldrich: CAS number: 67-68-5) with dilutions in the extracellular solution to obtain the final concentrations. The final DMSO concentration in hERG experiments was 0.1% and in Nav1.5 and Cav1.2, it was 0.3% (*v*/*v*). Samples were first stored at freezing, −80 °C, for later analysis.

### 2.2. Ion-Channel Screening of hERG, Nav1.5, and Cav1.2

The method applied for ion-channel IC_50_ was similar to our earlier study [47]. Ionic currents were evaluated in the whole-cell configuration using the patch-clamp method, manual patch-clamp (I_kr_ in hERG-transfected HEK293 cells), and a SyncroPatch^®^ 384 PE automated patch-clamp platform (Nav1.5 and Cav1.2—transfected CHO cells) (Nanion Technologies GmbH, Munich, Germany). All studies were conducted at room temperature (~23 °C).

Manual hERG Test Voltage Protocol: the hERG current was elicited from a holding potential of −80 mV by a voltage step to +60 mV for 2000 ms, followed by repolarization to −40 mV. This pattern was repeated at a rate of 0.07 Hz. Peak hERG current was measured at −40 mV. Before each test pulse, a short pulse (0.5 s), from the holding potential to −60 mV, was given to determine the leak current. The effect of loperamide was evaluated after 5 min of drug application. One to three concentrations of the drug were tested per cell (applied cumulatively).

SyncroPatch^®^ 384 PE Nav1.5 Test Voltage Protocol: Nav1.5-mediated current was measured using stimulus voltage patterns with fixed amplitudes (−5 mV) repeated at 10 s intervals from a holding potential of −110 mV. Different test concentrations of loperamide were given for ~4 min.

SyncroPatch^®^ 384 PE Cav1.2 Test Voltage Protocol: Cav1.2-mediated current was measured using stimulus voltage patterns with fixed amplitudes (+10 mV for 100 ms) repeated at 20 s intervals from a holding potential of −70 mV. Application of loperamide, verapamil (positive control), and solvent controls was 5 min, followed by 2 min application of 100 µM verapamil to reach a full block of Cav1.2-mediated current.

Data analysis of three ion channels: Patch clamp data were acquired and analyzed using Pulse and Pulsefit (version 8.30; HEKA), DataAccess (Bruxton), Igor (version 3.13; Wavemetrics) for the manual patch-clamp, and the SP384PE system operation software DataControl (Nanion Technologies GmbH, Munich, Germany) for the automated patch-clamp. The decrease in current amplitude was used to calculate the percent block relative to control baseline conditions:% Block = (1 − I_TailAmplitude_/I_Baseline_) × 100%

For Cav1.2, the data were corrected for rundown: %Block = 100% − ((%Block − %PC) × (100%/(%VC − %PC)),
where %VC and %PC were the mean values of the current block with the vehicle and positive controls, respectively.

Concentration/response relationships were calculated using non-linear regression fits using individual data points (GraphPad Prism 9.5). The half-maximal inhibiting concentration (IC_50_) was calculated by the fitting routine.
Y = Bottom + (Top − Bottom)/(1 + (IC_50_/X)^HillSlope).

### 2.3. *In Silico* Modeling

In order to attempt to assess the potential proarrhythmic effects of loperamide, we used the internally generated in vitro IC_50_ data for the three major ion channels [hERG (I_Kr_), Nav1.5 (I_Na_), and Ca (I_Ca_)] (Table 1) to simulate action potential changes in silico. The IC_50_ data were employed in an in-house developed in silico model (named Easy Systematic Channel Affinity Proarrhythmia Evaluation In silico Tool, ESCAPE-IT (Version 1.0)) that was built around the Virtual Assay drug screening software (Version 3.2) from the Computational Department of Oxford University, which is based on the O’Hara Rudy action potential algorithm (https://www.cs.ox.ac.uk/ccs/virtual-assay/) (accessed on 1 April 2023).

The ESCAPE-IT tool enables systematic proarrhythmic assessment at clinically efficacious exposures and multiples above using the Virtual Assay to simulate the compound’s behavior on a population of cardiac cells. The ESCAPE-IT tool automates post-processing and data visualization and facilitates the assessment of proarrhythmic development. The input parameters used for simulations in both models are presented in Table 1. The method applied in this study is similar to a paper published by other researchers [48,49,50].

Within the Virtual Assay, we started with an in silico cell population of 500 cells by randomly varying the individual ion-channel parameters within the algorithm [49]. Subsequently, a so-called “calibration step” was performed within the system to select cells only when they represent appropriate action potential physiology [48]. Following this, the settings of 315 cells were found to be physiologically relevant and useful for further in silico proarrhythmic modeling (consequently, 185 non-physiologically relevant cells were excluded). The physiological prolongation of APD_90_ was defined based on the upper limit APD_90_ (upper limit of ~440 ms of the baseline values) and compared with the value of negative control of metoprolol (at 30× FTPC: APD_90_ at 418 ms)**.** Proarrhythmic liability was assessed using ESCAPE-IT, applying the IC_50_ responses of loperamide and hydroxyzine in both non-calibrated and calibrated settings in the selected 315 cell population (See Appendix A). The simulated concentration was increased gradually until the first proarrhythmic event was observed. For that purpose, we tested loperamide at 200×, 400×, 600×, 800×, 1000×, 1200×, 1400×, and 1600× its maximum human-free therapeutic plasma concentration (FTPC) of 0.25 nM, and hydroxyzine at 1×, 2×, and 3× its human FTPC of 13 nM. In the present study, we took one example of hydroxyzine to mimic the clinical misuse of loperamide in combination with other medications [51]. The FTPC of hydroxyzine was taken from the Elseviers’ PharmaPendium DATABASE regulatory submission data search engine: www.pharmapendium.com (accessed on 1 April 2023). Simulations were run for loperamide alone and in combination with hydroxyzine (Table 1). For this in silico modeling purpose to see if there are any synergic effects on APs, we only applied IC_50_ values of hERG (I_Kr_), Na_v_1.5 (I_Na_), and Ca_v_1.2 (I_Ca_) for the stimulation [52]. For metoprolol, a CiPA* negative control drug without QT prolongation and TdP risk [53], IC_50_ values of 145 µM, 630 µM, and 3280 µM for hERG (I_Kr_), Na_v_1.5 (I_Na_), and Ca_v_1.2 (I_Ca_), and an FTPC of 1.8 µM [54] were used for the stimulation as a comparison and used to define the cut-off value of relevant physiological changes since it was used as a negative control in the Comprehensive in Vitro Proarrhythmic Assay (CiPA*) study [55].

### 2.4. Isolated Arterially Perfused Rabbit Ventricular Wedge and Electrophysiological Recordings

The preparation of the isolated rabbit ventricular wedge and its electrophysiological recordings were described in detail in our earlier papers [56,57] and similar to other published papers [58,59]. The small ventricular preparation (±1.5 cm × 2 to 3 cm) was placed in a tissue bath and arterially perfused with Tyrode’s solution containing 4 mM K^+^ buffer with 95% O_2_ and 5% CO_2_ (temperature: 35.7 ± 0.1 °C, perfusion pressure: 30 to 50 mmHg).

Transmembrane action potentials in wedge preparations were recorded simultaneously from endocardial sites by use of a separate intracellular floating microelectrode and a transmural electrocardiogram (ECG) was recorded concurrently in all experiments. The following ECG parameters or incidence of cardiac arrhythmias were measured or noted: QT interval, defined as the time from the onset of the QRS complex to the point at which the final downslope of the T wave crossed the isoelectric line; JT interval (QT-QRS); Tp–Te interval (time from the peak to the end of the T wave, a measure of transmural dispersion of repolarization [TDR]); Tp–Te/QT ratio (rTp); QRS duration; QRS rate dependency; iCEB (QT/QRS or JT/QRS ratio); occurrence of phenomena dependent on phase 2 EAD (early afterdepolarization); inexcitability (preparation unable to follow the electrical stimulation); ventricular tachycardia (VT); and ventricular fibrillation (VF). In addition, a score predicting the TdP risk in humans was calculated as previously defined [56,58,60].

The study protocol employed was similar to our previously established protocol (pacing at 0.5 Hz and then 1 min pacing at 2 Hz at the end of each dose [56]). Loperamide was continuously perfused at five increasing concentrations of 0.1 µM, 0.3 µM, 1 µM, 3 µM, and 10 µM (n = 5) or vehicle (n = 5).

### 2.5. Anesthetized Guinea Pigs

Fourteen female guinea pigs (Dunkin-Hartley Charles River Germany) weighing 680–840 g were randomly assigned to the vehicle or loperamide treatment groups using a protocol similar to our earlier paper [61], and to other published papers [62,63]. Guinea pigs were anesthetized with sodium pentobarbital (60 mg/kg i.p.), maintained by continuous i.v. infusion of ~6 mg/kg/h sodium pentobarbital, and ventilated. Two carotid arteries and two jugular veins were cannulated for monitoring arterial blood pressure, collection of blood samples, and intravenous administration. Surface ECG (Lead II) and blood pressure were recorded continuously and analyzed online and offline with Notocord^®^ software.

Intravenous infusions of incremental doses of loperamide (or equivalent vehicle volumes) were administered to two separate groups of animals with each dose infused over 5 min at 15 min intervals at incremental doses of 0.16, 0.31, 0.63, 1.25, 2.5, and 5 mg/kg (total cumulative dose = 9.85 mg/kg; n = 7) or vehicle (0.5, 0.5, 0.5, 0.5, 0.5, and 0.5 mL/kg; n = 7). Arterial blood samples (0.3 mL) were drawn at the end of each dose infusion and blood samples from animals dosed with loperamide were collected in Eppendorf tubes containing 10 μL of heparin (1000 I.U./mL) and centrifuged (at 8000 rpm for 2 min) immediately after collection. Plasma samples were stored frozen for later analysis of loperamide concentrations. Averages of 10 consecutive beats before the selected time points (at 2 and 5 min after the onset of each dose infusion) were taken for validation of the ECG intervals and statistical analysis.

For statistical analysis of data from the isolated rabbit wedge and anesthetized guinea pigs: changes versus the corresponding baseline values (in actual units) in the loperamide group were compared with those in the vehicle group using the Wilcoxon–Mann–Whitney test (The R Project for Statistical Computing Version R software = 2.15.2, Version Cardiovascular Algorithm = 0.0.29). *p* values < 0.05 were considered statistically significant.

### 2.6. Plasma Protein Binding Determinations

Plasma protein binding of loperamide in human and guinea pig plasma was determined at several concentrations encompassing therapeutic concentrations and concentrations likely associated with abuse and overdose using pooled blood samples from guinea pigs (Charles River, Dunkin Hartley) and humans (3 males) in K_2_EDTA. Human whole blood donors were under 55 years of age, fasted for a minimum of 12 h, were medication free for a minimum of two weeks, and consumed no smoke and alcohol within 24 h of blood donation. Plasma was prepared and stored at −80 °C prior to use.

Protein binding was determined by equilibrium dialysis using the Dianorm system. Blank plasma was spiked with two hundred-fold concentrated stock solutions of loperamide (0.2, 6, 20, 40, 80 µg/mL in DMSO) to final concentrations of 1, 30, 100, 200, and 400 ng/mL in plasma, respectively (0.5% DMSO final concentration), at room temperature. Fortified plasma samples were subjected to dialysis against 0.067 M phosphate buffer at pH 7.21, for 4 h at 37 °C in a Dianorm system with identical macro-1 Teflon cells and Spectrapor^®^RC 2 dialysis membranes (MW cut-off of 12–14 kDa). After dialysis, the contents of the two compartments of the dialysis cells were collected separately. The contents of the buffer compartments were collected into test tubes containing 0.2 mL 10% bovine serum albumin solution to minimize the potential for non-specific binding of test compounds to the collection tubes. The volumes of each of the buffer compartments were determined by weighing. At the time of dose administration, samples were collected for determination of the start concentration of test compound. The reference compounds propranolol (in triplicate) and warfarin (in duplicate) were both tested at 2 µg/mL. Samples were stored at −20 °C until analysis using liquid chromatography coupled with mass spectrometry (LC-MS/MS).

### 2.7. Analysis of Well or Bath Loperamide Concentrations

For the ion-channel studies, graded concentrations of loperamide in experimental buffer were perfused through the apparatus in a separate series of experiments (with cells present but without making electrophysiological measurements) and samples were taken for determining drug concentration. The samples taken from the wedge study and anesthetized guinea pig study were also collected and analyzed by using liquid chromatography coupled with mass spectrometry.

### 2.8. Cardiac Safety Margin Calculations

The maximum recommended adult daily dose of loperamide for acute diarrhea is 8 mg/day for over-the-counter loperamide and 16 mg/day for prescription loperamide. The exposure at the higher daily therapeutic dose was used for safety margin calculations. In a bioequivalence study, 16 mg of Janssen loperamide capsules were administered orally to fasting subjects, and a mean total Therapeutic Plasma Concentration TPC (C_max_) of 3.98 ng/mL was obtained [64]; this C_max_ was higher than those reported in Pharmapendium (https://www.pharmapendium.com, accessed on 1 April 2023). These values, along with the calculated unbound concentrations (free therapeutic plasma concentration: FTPC at 0.119 ng/mL = 0.25 nM) based on 3% human protein binding, were used to calculate margins for the various findings at nominal concentrations used in in vitro experiments (isolated cells in patch clamp and isolated rabbit ventricular wedge preparation) as well as at plasma concentrations achieved in the anesthetized guinea pig experiments. Guinea pig total plasma concentrations were converted to free concentrations using calculated plasma protein binding for guinea pig plasma and compared with human free plasma concentrations based on plasma protein binding determinations in human plasma. Given the lack of protein in the in vitro studies, nominal in vitro concentrations were compared with peak unbound plasma concentrations in humans.

## 3. Results

### 3.1. Effects of Loperamide on Cardiac Ion Channels

The effects (IC_50_ values) of loperamide on hERG-mediated I_Kr_, a Nav1.5-mediated sodium current (I_Na_) (n = 28–35 each concentration), and a Ca_v_1.2-mediated calcium current (I_Ca_) (n = 10–12 each concentration) are presented in Figure 1.

Loperamide inhibited hERG-mediated I_Kr_ with an IC_50_ value of 0.39 µM (n = 5 each concentration), a Nav1.5-mediated sodium current (I_Na_) with an IC_50_ value of 0.526 µM, and a Cav1.2-mediated calcium current (I_Ca_) with an IC_50_ value of 4.09 µM (Figure 1). Noticeably, these IC_50_ values are ≥1560-fold (×) its human FPTC.

### 3.2. *In Silico* Modeling of Loperamide Effects on Human Cardiac Action Potentials

In silico, the modeling parameters (Table 1, presented in Section 3.1) were simulated based on their IC_50_ values on the three cardiac ion channels. The simulated evolution of APs with loperamide alone at increasing concentrations, relative to its human FTPC, is summarized as follows: loperamide had no physiologically relevant effects on APD_90_ prolongation up to 600-fold (×) its human FTPC (APD_90_ = 417 ms), similar to that with the negative control metoprolol at 30× FTPC = 418 msec, and produced the physiologically relevant prolongation of AP duration from ×800 and elicited early afterdepolarization (EAD) from 1400× FTPC. At ≥800× FTPC, loperamide also increased the triangulation of the AP (APD_90_-APD_40_), an indicator of the shape change in AP and proarrhythmic biomarkers [65] (Figure 2).

In silico action potentials are presented in Figure 2 (Individual graphs are presented with a *y*-axis in millivolts and *x*-axis time in ms. In addition, the overall figure shows multiples of loperamide in the *x*-axis and hydroxyzine in the *y*-axis). Interestingly, the cotreatment of different doses of loperamide with different doses of hydroxyzine (expressed as different concentrations from 1× to 3× the FTPC) gradually reduced safety margins: AP duration reduced from 600× FTPC to 200× FTPC, while the safety margin for induction of EAD was reduced from 1200× FTPC to 800× FTPC. Hydroxyzine inhibits multiple ion channels such as hERG (IC_50_ = 0.39 µM), Na_v_1.5 (IC_50_ = 13.3 µM), and Ca_v_1.2 (IC_50_ = 8.6 µM) and has been reported to prolong QT interval and produce TdP in man [52].

The parameters obtained from the patch clamp data for loperamide, alone and in combination with hydroxyzine and metoprolol (a negative control) on cardiac electrophysiology, are provided in detail in Appendix A.

### 3.3. Isolated Arterially Perfused Rabbit Left Ventricular Wedge

Interestingly, loperamide did not mimic drug-induced long QT syndrome in the isolated rabbit wedge assay (a sensitive QT and TdP biomarker assay): QT interval and JT-interval were not prolonged and the TdP score, transmural dispersion (Tp-Te and rTp-Te), and JTp were decreased rather than increased (Appendix A).

Relative to the vehicle (n = 6), loperamide (n = 6) at 0.1 µM, 0.3 µM, and 1 µM did not significantly change the QT interval or JT-interval (Figure 3). At 3 µM [~12,000-fold (×) FTPC] and 10 µM (~40,000× FTPC), loperamide significantly shortened the QT- and JT-intervals. Loperamide did not affect QRS duration at 0.1 µM but significantly increased QRS starting at 0.3 µM (+6%, +18%, and +86% of baseline at 0.3 µM, 1 µM, and 3 µM) vs. 0% of baseline with the vehicle (*p* < 0.05) (Figure 3). Although loperamide did not significantly affect the index of cardio-electrophysiological (a biomarker of risk) balance (iCEB) at 0.1 µM and 0.3 µM, iCEB was significantly reduced at 1 and 3 µM (~4000× FTPC) (Figure 3). Loperamide at 0.1 µM to 10 µM, did not elicit EADs, TdP, or VF in rabbit ventricular wedge preparations. However, loperamide elicited cardiac ectopic beats at 3 µM (~12,000× FTPC), and non-TdP-like ventricular tachycardia (VT) at 10 µM (~40,000× FTPC) in all five preparations, and induced inexcitability in one out of five preparations at 10 µM. Examples of cardiac arrhythmias at 3 and 10 µM are presented in Figure 3D,E. At the end of perfusion with the highest nominal loperamide concentration (10 µM), the median concentration of loperamide in the perfusate near the wedge/tissue preparation was 6.23 µM.

### 3.4. Anesthetized Guinea Pigs

The median (minimum/maximum) plasma concentrations of loperamide at the end of infusions of 0.16, 0.31, 0.63, 1.25, 2.5, and 5 mg/kg were 96 (59/137), 155 (133/266), 334 (290/490), 727 (608/1300), 2100 (1600/3310), and 9080 (1290/19,300) ng/mL, respectively [data were in Median (maximal/minimal)]. A fraction unbound of 5% in the guinea pig plasma (as determined in the current studies) was used for the calculation of free plasma levels and for determining multiples of human FTPC.

Loperamide had no significant effect on ECG parameters through a dose of 1.25 mg/kg i.v., except for a slight increase in PQ interval at 1.25 mg/kg (free plasma level of 36 ng/mL; ~304× human FTPC). Starting at 2.5 to 5 mg/kg (free plasma concentration of 105 to 454 ng/mL, respectively; 879× to 3802× human FTPC, respectively; Figure 4), loperamide significantly changed ECG parameters: increases in QRS-duration and PQ-interval with lesser increases in QTcB-interval (QT corrected with Bazett: QTcB = QT/RR) and JT-interval (Figure 4). The prolongation of the JT-interval was larger than that of the QTcB, suggestive of potassium channel blocking effects. Additionally, loperamide at 2.5 mg/kg i.v. significantly reduced the index of the cardio-electrophysiological balance (iCEB = QTcB/QRS ratio). The effects on heart rate, arterial blood pressure, and other ECG parameters are provided in Appendix A. The detailed effects on heart rate and arterial blood pressure and QT-interval are provided in Appendix A.

Abnormal ECG complexes, which are indicative of conduction disturbances (axis deviation, BBB, P-on-T), were only observed in loperamide-treated animals at 2.5 and 5 mg/kg (free plasma level = 105 and 454 ng/mL, respectively; 879× and 3802× human FTPC, respectively). At 2.5 mg/kg, abnormal ECG complexes and conduction disturbances were observed in some animals. At 5 mg/kg, abnormal ECG complexes and conduction disturbances (axis deviation, BBB, 2nd AVB, 3rd AVB) were observed in six of seven animals and cardiac arrest occurred in four of seven animals. Examples of ECG recording in guinea pigs are represented in Figure 4J.

As expected, dofetilide (0.02 mg/kg/i.v. infused over 1 min), administered 15 min after the onset of the last infusion of the vehicle, reduced heart rate (median decrease of −15% 5 min after infusion) and prolonged QT and QTcB intervals (median increases of +22% and +12%, respectively) without significantly changing QRS duration.

### 3.5. Plasma Protein Binding

The percentage of unbound loperamide at 30, 100, 200, and 400 ng/mL in human plasma was 2.76%, 2.97%, 3.05%, and 3.23% respectively, and in the guinea pig plasma, it was 4.67%, 5.04%, 5.36%, and 6.04% respectively. Note that the percent of unbound loperamide at 1 ng/mL in human and guinea pig plasma could not be determined because loperamide concentrations in the buffer (unbound) were below the minimum level of quantitation. Given the minimal concentration dependence for the fraction unbound of loperamide in human and guinea pig plasma, the fraction unbound in human and guinea pig plasma at all concentrations was assumed to be 3% and 5%, respectively. These values were used to calculate free plasma concentrations in each species as well as multiples of human FTPC. Percent unbound for the control compounds, propranolol (mean ± SD, n = 3 for human: 14.6 ± 1.2; guinea pig: 7.36 ± 0.12) and warfarin (mean of n = 2 for human: 0.593, guinea pig: 2.14), were within the expected range.

### 3.6. Safety Margin Calculations

The summary of safety margin ratios calculated from data in different preclinical assays based on its maximum human therapeutic free plasma concentration (FTPC) is presented in Table 2.

## 4. Discussion

The recent increases in the abuse and intentional overdose of the antidiarrheal drug, loperamide, which acts via an opiate mechanism of action, has been accompanied, in many cases, by severe cardiac toxicity consisting of pronounced QT prolongation, QRS widening, AV blockade, and associated life-threatening ventricular arrhythmia, such as TdP and other forms of ventricular tachycardia and ventricular fibrillation [14,34,35]. The present nonclinical in vitro, in vivo, and in silico studies with loperamide provide a mechanistic basis for the cardiac electrophysiological toxicity of loperamide at extreme overdoses associated with abuse. At extremely high exposure levels, as in these abuse cases in men, loperamide can cause cardiac ventricular repolarization shortening and conduction slowing (in vitro at ≥1200× human FTPC), and further cause ventricular tachycardia-like activity in the rabbit wedge assay. In anesthetized guinea pigs, loperamide largely slowed conduction time (QRS-duration), increased PQ-interval and QTcB-interval at a free plasma level of 879× its human FTPC, and subsequently, it caused Type II/III AV block, or even cardiac arrest in anesthetized guinea pigs (at >879× or 3802× its human FTPC). The slowing of conduction, the shortening of ventricular repolarization time, and the consequent proarrhythmic risk are likely mediated by loperamide’s I_Na_ channel-blocking activity. Our preclinical results are also consistent with certain other features in reported clinical cases of abuse and extreme overdose: conduction defects (QRS widening; bundle branch block) and QT-prolongation, which in many cases were associated with polymorphic ventricular tachycardia, sometimes labeled as TdP [15,34,66].

### 4.1. Safety Margins over Loperamide’s Free Therapeutic Peak Concentration (FTPC)

The maximum recommended adult dose of loperamide in acute diarrhea is 16 mg/day. In a bioequivalence study, 16 mg of loperamide capsules administered to fasting subjects resulted in a mean C_max_ of 3.98 ng/mL [64], which was used to estimate maximum therapeutic exposures. Unbound therapeutic C_max_ or FTPC (0.12 ng/mL; 0.25 nM) based on 97% human plasma protein binding was used for safety margin calculations based on in vitro systems using a protein-free test medium. The margins for the effects in the various models, as well as maximum no-effect exposures, are presented in Table 2. Overall, the safety margins for the effects on cardiac repolarization and conduction in nonclinical models in comparison with maximum therapeutic exposures are very large: 1560-fold for the hERG IC_50_, (186-fold for the hERG IC_10_) 400-fold for the no-effect concentration in the rabbit ventricular wedge, and 304-fold for the highest no electrophysiological effect plasma concentration in the anesthetized guinea pig. Even using the lowest published hERG IC_50_ of 33 nM [39], the margin of 132-fold is still quite large. When analyzed according to an “Integrated Risk Assessment” proposed by ICH S7B Guidelines [67], the electrophysiological in vitro and in vivo studies suggest no potential of loperamide, at recommended dosing levels, to prolong the QT interval or affect cardiac conduction in humans. This is consistent with the lack of reported cardiac toxicity over the more than 30-year history of the therapeutic use of loperamide at recommended dose levels [2,7,8].

### 4.2. Relationship between Exposures for Nonclinical Cardiotoxicity and Plasma Levels Associated with Intentional Overdose

Although plasma levels of loperamide were measured in a portion of published cases, these samples were drawn at varying intervals after an arrhythmia incidence and the ingestion of loperamide (on presentation up to 3 to 6 days later). Thus, the lack of systematic plasma concentration measurements associated with the reported cases of loperamide cardiotoxicity makes comparisons to nonclinical findings more difficult. Indeed, it has also been acknowledged that substance abusers are often not forthcoming with important metadata information (other drugs taken, etc.,) upon questioning [68]. Nonetheless, we used an available range of reported human plasma levels for comparison with our nonclinical test system findings: lower total PC of 33 ng/mL (FPC = 1 ng/mL or 2.1 nM; ~8.4× FTPC) [18] to upper toxic PCs of 210 to 890 ng/mL (FPC = 6.3 to 30 ng/mL; ~52.9 to 252× FTPC) [24,69]. Given that the half-life of the drug increases from 9–14 h to >40 h at doses ≥16 mg/day [70], and the real FPC could be much higher due to potential drug accumulation when patients take, for example, loperamide >100 pills/day for >1 year. The exposures associated with cardiac electrophysiological toxicity in our nonclinical studies are generally well above this estimated range of exposures associated with reported loperamide toxicity in humans. In addition to the lack of appropriate metadata mentioned above, this difference may be due to several other different factors as listed below:
(1)Concomitant medications as well as underlying conditions (e.g., hypokalemia) may alter the sensitivity to loperamide’s actions. Concomitant medications in reported cases included sotalol, nintedanib, methadone, amitriptyline, fluoxetine, clonazepam, and alprazolam, etc., [37], which are known to have direct effects on cardiac ion-channels (e.g., hERG) and may induce long QT and TdP in humans by themselves. In addition, drug misusers often took P-glycoprotein inhibitors to increase brain levels of loperamide to achieve an opiate “high” (e.g., ketoconazole, fluoxetine, citalopram, omeprazole, quinine, verapamil, erythromycin, Hydroxyzine). Many of these P-glycoprotein inhibitors also directly affect ventricular depolarization and repolarization (QT-interval) [12,71].(2)As previously mentioned, the available estimates of loperamide exposure associated with overdose cases were not systematically collected and were taken at varying times after loperamide ingestion or the presentation of cardiac toxicity, suggesting the actual peak concentrations may have been much higher at the time of events.(3)Loperamide, a substrate for P-glycoprotein [4,72], may saturate this transporter at toxic concentrations or in the presence of other drugs that inhibit P-glycoprotein [73,74], resulting in significantly higher plasma levels. As this transporter also excludes loperamide from cardiac cells, intracellular levels may be significantly higher [75].(4)In vitro or in silico studies do not account for the potential pharmacological activity of metabolites. Two human metabolites of loperamide (N-desmethyl loperamide and N-hydroxymethyl-mono-desmethylloperamide) are generated at levels greater than the parent drug. In two overdose cases, desmethyl loperamide concentrations were 5- to 8-fold the parent levels [28]. Desmethylloperamide has been shown to inhibit hERG with an IC_50_ of 245 nM and I_Na_ with an IC_50_ of 483 nM [76] and therefore may accentuate the ion channel effects of loperamide itself in the heart.(5)Compound solubility in aqueous buffer and variable drug adherence to the perfusate tubing and apparatus used in in vitro studies potentially limit actual testing exposure, leading to potential underestimates of potency and the overestimating of safety margins. For the internal hERG assessments, the recovery of loperamide in the perfusate ranged from 43 to 60%, and comparable recovery was found in the isolated rabbit ventricular wedge experiments.(6)Significant physiological consequences of ion-channel inhibition (ie., effects on conduction and repolarization) have been documented to occur at much lower levels of channel inhibition (e.g., IC_10_ to IC_20_) for both I_Kr_ [77] and for I_Na_ [38,78]. The safety margin was 186-fold when we applied the IC_10_ value of hERG.

### 4.3. Potential Mechanisms of Cardiac Arrhythmias Associated with Abuse and Extreme Overdose of Loperamide

The majority of the reported cases of loperamide cardiac toxicity were associated with greatly widened QRS duration (up to 200 ms), conduction defects, and marked QTc-prolongation (up to 704 ms), in some cases associated with TdP-*like* VT. Cardiac arrhythmias (TdP, VT, cardiac arrest) were often associated with QRS-prolongation or Brugada syndrome ECG features [14,33,79].

The exact mechanisms of cardiac arrhythmias associated with abuse and extreme overdoses of loperamide are unclear but are likely linked to loperamide’s inhibitory effects on cardiac ion-channels at high concentrations. The inhibition of hERG is the most common mechanism for drug-induced QT prolongation and, in some situations, can lead to the induction of TdP [42,43]. Our preclinical studies have confirmed that loperamide inhibits hERG-mediated I_Kr_ in HEK293 cells with an IC_50_ of 390 nM (based on nominal concentrations). In other published studies, loperamide was reported to inhibit hERG at lower IC_50_ values from 33 to 89 nM [39,40,41] under different study conditions and measurement protocols. The differences in potency determinations may be due to different voltage pulsing protocols, test system temperatures, compound recovery (actual concentrations found in the bath), and other experimental differences [55,80,81,82]. Variability in IC_50_ determinations (≤3 fold), with the same manual patch-clamp protocol in different labs, for drugs with hERG inhibitory properties is well documented [83]. Thus, these values should be considered comparable and qualitatively consistent with loperamide’s electrocardiographic actions.

In our studies, loperamide also inhibited I_Na_ with an IC_50_ of 526 nM (comparable to its IC_50_ for hERG). Although loperamide was shown to inhibit I_Na_ with an IC_50_ of 2900 nM using a high-throughput screening system [38], a conventional manual patch clamp study determined an IC_50_ for I_Na_ of 239 nM [39], comparable to our results. The concurrent inhibition of I_Na_ may offset repolarization prolongation caused by hERG inhibition or even result in QT shortening, as well as QRS widening, conduction disturbances, and conduction slowing (PR-interval prolongation, AV block, bundle branch block), as was shown in the rabbit wedge study and in anesthetized guinea pigs. Our findings in anesthetized guinea pigs are similar to a recently reported study of loperamide within the same species under anesthesia [84]. This profile is also consistent with a study in isolated swine cardiomyocytes that showed that loperamide (10 and 100 nM; 40× and 400× human FTPC) prolonged action potential duration but had no effect on APD at 1000 nM [40]. Moreover, at this concentration, there was a trend for a decrease in V_max_, a reflection of I_Na_ inhibition on action potential upstroke. Overall, this electrophysiological profile, consisting of conduction slowing and the shortening of repolarization, may predispose to ventricular tachycardia [56]. Non-TdP-like VT was noted in the isolated rabbit ventricular wedge preparation at very high concentrations. Marked QRS widening and VT (often labeled TdP) have also been noted in many reported cases of loperamide overdose [14].

The IC_50_ of loperamide for I_Ca_ was relatively high (IC_50_ = 4091 nM), and thus, its inhibition seems less likely to contribute to or offset the proarrhythmic potential for loperamide due to its much more potent hERG and I_Na_ inhibition. However, at extremely high concentrations, I_Ca_ inhibition in cardiomyocytes could contribute towards the slowing of atrioventricular conduction and cardiac arrest observed in the anesthetized guinea pig and observed in many human overdose cases. Interestingly, the index of cardio-electrophysiological balance (iCEB = QT/QRS ratio), as a novel risk marker for predicting malignant ventricular arrhythmia, was significantly reduced in both the isolated rabbit model and in anesthetized guinea pigs. We have reported that significant increases in iCEB is associated with long QT and TdP, while a significant reduction in iCEB is associated with non-TdP-like VT/VF [85,86]. iCEB is significantly reduced in patients with Brugada syndrome and with drugs that block the I_Na_ channel [86,87]. In the rabbit wedge study, cardiac arrhythmias (ventricular couplets and non-TdP-like VT) at 3 and 10 µM loperamide were associated with a significant reduction in iCEB values via increases in QRS-duration and the shortening of the QT/JT-interval. Similarly, in anesthetized guinea pigs, loperamide significantly reduced iCEB largely via increasing QRS duration. The effect of loperamide on iCEB in the present study is similar to that of flecainide in the rabbit wedge model and in humans [85,86].

### 4.4. In Silico Assessment of Loperamide Cardiotoxicity at Concentrations Associated with Overdose

In silico models of the human ventricular action potential have been significantly refined and evaluated for the estimation of the clinical cardiac electrophysiological risk of drugs based on the integration of their effects on multiple human cardiac ion channels [49,88,89,90]. The application of these models has been systematically explored for predicting human proarrhythmic risk as part of the Comprehensive in Vitro Proarrhythmic Assay (CiPA) [91,92,93]. Prototype studies of many drugs with varying cardiac electrophysiological actions and proarrhythmic risks have been published [93,94,95]. Our in silico evaluations at up to 600 times of FTPC of loperamide alone showed no significant changes in APD’s, consistent with our integrated risk assessment based on in vitro (600-fold in wedge) and in vivo (304-fold in the guinea pig) data. At over 1200× FTPC, loperamide did prolong the APD and induced early depolarizations (EADs), consistent with proarrhythmic risk. The findings are similar to a previously reported in silico assessment of loperamide on human cardiac action potentials and proarrhythmic potential at therapeutic and overdose concentrations [41]. Interestingly, this in silico analysis, which included the modeling of action potential upstroke, showed a concurrent reduction in maximum upstroke velocity (V_max_), consistent with I_Na_ inhibition.

When we attempted to mimic clinical conditions where loperamide is used in combination with another drug that blocks hERG and has proarrhythmic potential, such as hydroxyzine, the safety margin of loperamide was greatly reduced (Figure 2). Hydroxyzine is an antihistamine that also inhibits hERG with an IC_50_ of 0.16 µM [96] and has also been used together with loperamide in the clinic. Based on its Free C_max_ of 0.013 µM, taken from the PharmaPendium website (www.pharmapendium.com) (accessed on 1 April 2023), the safety margin over its IC_50_ is about 12× its human FTPC. A combination treatment of loperamide with hydroxyzine would be expected to significantly reduce the safety margin in a clinical setting. Indeed, in in silico modeling, the concentration of loperamide to produce an abnormal action potential interval and arrhythmia was much lower than in loperamide treatment alone (Figure 2).

## 5. Conclusions

Our studies and overall analysis of in vitro and in vivo nonclinical findings for loperamide fully support the high safety margins of loperamide and its well-established clinical safety profile, when used within its recommended therapeutic dose range for the treatment of acute diarrhea. However, considering the reported cases of cardiac arrhythmia, associated with extreme systemic exposure following the intentional overdose/abuse of loperamide, our data provide evidence of likely pharmacological mechanisms for the cardiac electrophysiological effects and ventricular proarrhythmia noted in these cases. Our nonclinical data indicate that loperamide at these high exposures would result in the slowing of conduction (AV blockade), wide QRS, reduction in iCEB, and non-TdP-like polymorphic VT/VF. These effects are a result of the inhibition of I_Na_. The reported cases of cardiac arrhythmia associated with the prolongation of QT-interval and incidence of apparent TdP with loperamide overdose are likely attributable to loperamide’s inhibition of the I_Kr_ (hERG) channel and potential additive effects when administered with other drugs that inhibit hERG and/or PGP (e.g., Sotalol, hydroxyzine).

## Figures and Tables

**Figure 1 biomolecules-13-01355-f001:**
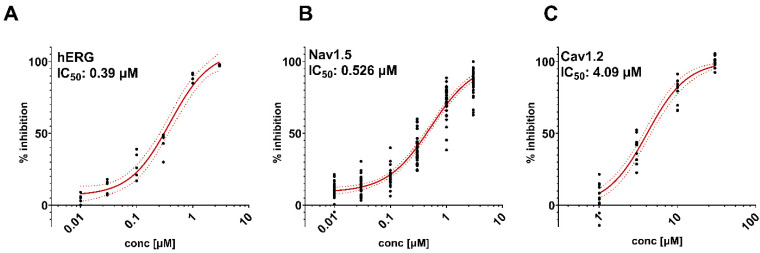
Effects of loperamide on IC_50_ values in 3 cell lines on hERG (n = 5 each concentration) (**A**), I_Na_. (n = 28–35 each concentration) (**B**), and I_Ca_ (n = 10–12 each concentration) (**C**) channels.

**Figure 2 biomolecules-13-01355-f002:**
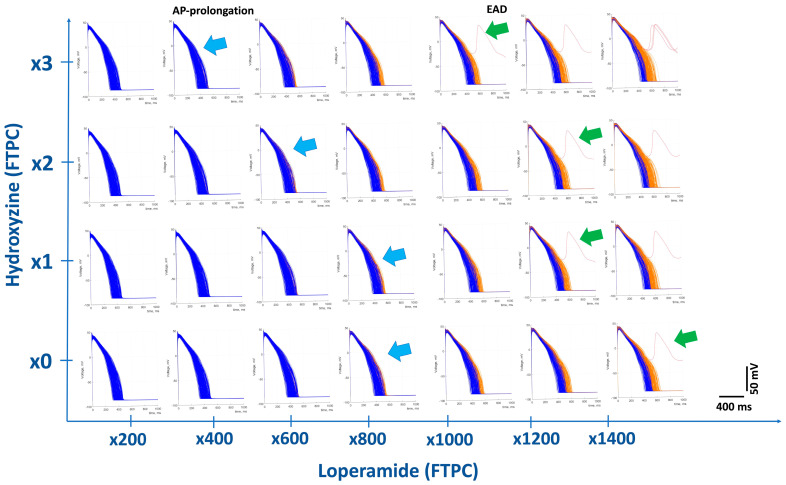
In silico effects of loperamide alone and in combination with hydroxyzine on simulated populations of action potentials. Modeled evolution of changes in cardiac action potentials (APs) and incidence of early afterdepolarization (EAD) of the cell population at different concentrations of loperamide in relation to its human FTPC using simulation parameters. The bottom row shows traces of the effects of loperamide alone; the upper rows show the effects of loperamide with concurrent hydroxyzine at 1×, 2×, and 3× its FTPC. Combination treatment with hydroxyzine reduced the margin of loperamide to prolong the action potential duration (AP) (from 800× to 400× FTPC) and to elicit early afterdepolarization (EAD) (from 1400× to 1000× FTPC), as noted by the left shifting arrows. Indeed, loperamide in combination with hydroxyzine synergistically prolonged APD_90_ (at higher multiples) and therefore reduced the safety margin of the risk for QT-prolongation.

**Figure 3 biomolecules-13-01355-f003:**
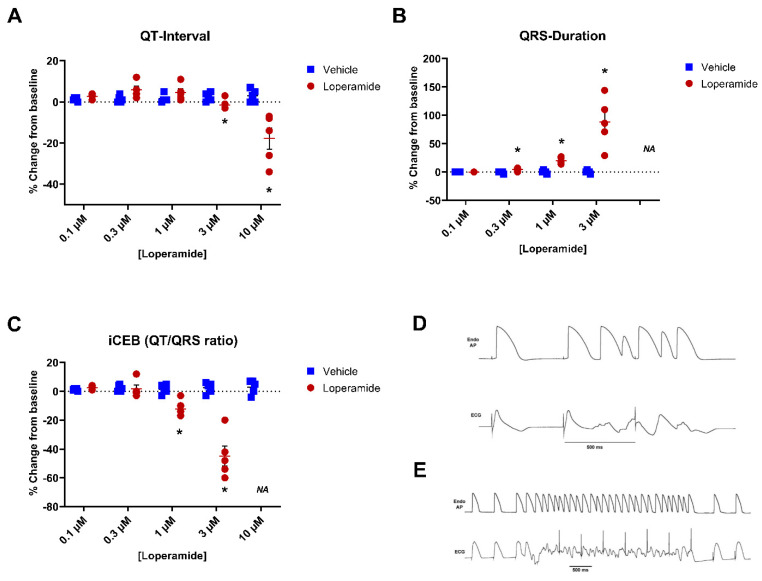
**Effects of loperamide on QT interval, QRS duration, and iCEB (QT/QRS ratio) in the isolated rabbit ventricular wedge.** There were no significant effects on QT interval up to 1 µM and iCEB up to 0.3 µM. Significant shortening in QT interval at 3 µM, Panel (**A**), and iCEB, Panel (**C**), at 1 and 3 µM and increases in QRS duration, Panel (**B**), starting at 0.3 µM. Panel (**D**) shows an example of ventricular couplets in a preparation exposed to loperamide at 3 µM). Panel (**E**) shows a non-TdP-like ventricular tachycardia (VT) in another preparation exposed to loperamide at 10 µM (24,800× FTPC). * *p* < 0.05 vs. solvent control.

**Figure 4 biomolecules-13-01355-f004:**
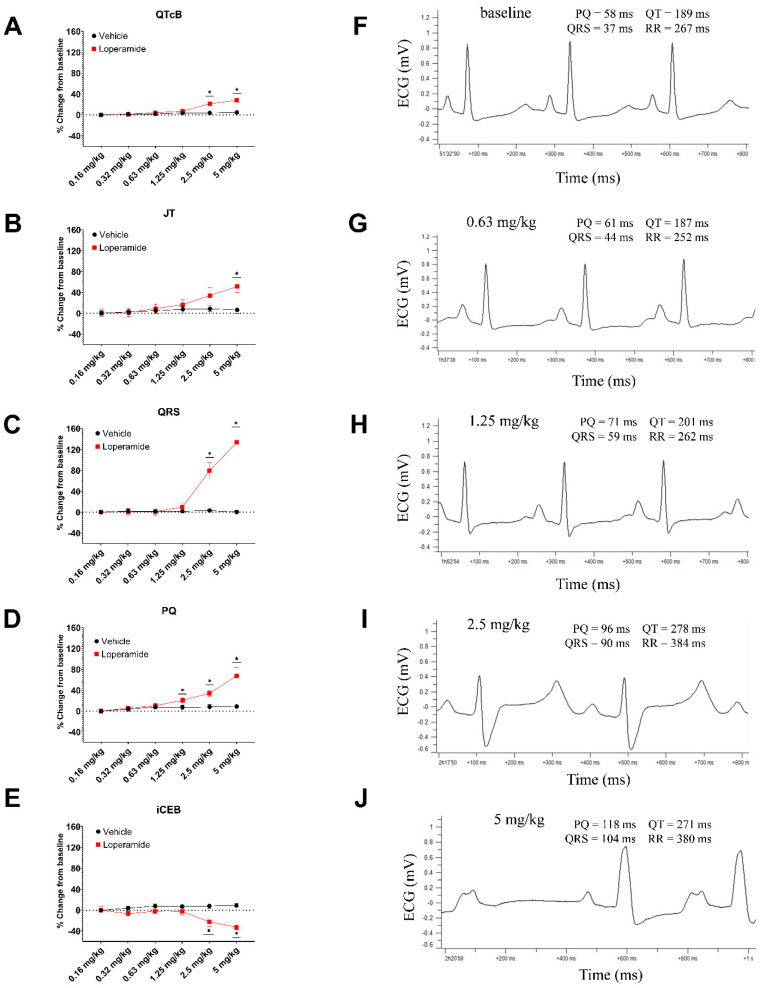
Effects of loperamide on QTcB interval (**A**), JT (**B**), QRS duration (**C**), PQ-interval (**D**), and iCEB (QT/QRS ratio) (**E**) in the anesthetized guinea pigs. (**F**,**G**): ECG recordings in the anesthetized guinea pig; Loperamide up to 0.63 mg/kg i.v. (**G**) had no significant effects on ECG parameters. Starting at 1.25 mg/kg i.v., (**H**) loperamide significantly changed ECG parameters: at 2.5 mg/kg i.v., loperamide caused AV-block type II (**I**) and AV-block type III at 5 mg/kg i.v. (**J**). * *p* < 0.05 vs. solvent control.

**Table 1 biomolecules-13-01355-t001:** The list of input modeling parameters, including limits of electrophysiological parameters and ion-channel IC_50_ values used to predict drug effects on the cardiac Action Potentials (APs).

Action Potential Biomarkers (Parameters) (Lower and Upper Limit)		
Parameter	Lower limit	Upper limit
Peak Voltage (mV)	10	55
RMP (mV)	−95	−80
APD 90 (ms)	180	440
APD 50 (ms)	110	350
APD 40 (ms)	85	320
Tri 90–40 (ms)	50	150
**Hydroxyzine**		
Ion-channel inhibited	IC_50_ (µM)	Hill coefficient.
hERG (I_Kr_)	0.39	1
Na_v_1.5 (I_Na_)	13.3	1
Ca_v_1.2 (I_Ca_)	8.6	1
**Loperamide**		
Ion-channel	IC_50_ (µM)	Hill coefficient
hERG (I_Kr_)	0.3897	1.21
Na_v_1.5 (I_Na_)	0.526	1.11
Ca_v_1.2 (I_Ca_)	4.084	1.64
**Free Therapeutic Plasma Concentration (FTPC)**		
Loperamide	0.25 nM	
Hydroxyzine	0.013 µM	

RMP: resting membrane potential, APD_40_, APD_50_, APD_90_: the duration of the action potential at 40%, 50%, and 90% repolarization.

**Table 2 biomolecules-13-01355-t002:** Safety margins of loperamide in preclinical models, relative to its maximum human free therapeutic plasma concentration (FTPC).

Test System	Parameter	Tested Dose	Margin
			X FTPC
I_Kr_ (hERG)	IC_50_	390 nM	1560
I_Na_	IC_50_	526 nM	2104
I_Ca_	IC_50_	4091 nM	16,364
In Silico Modelling	NE on APs	150 nM	600
	Significant effects on APs	200 nM	800
Rabbit ventricular wedge	NE	100 nM	400
	↑ QRS	300 nM	1200
	↓ iCEB	1000 nM	4000
	Cardiac arrhythmias	3000 nM	12,000
Anesthetized guinea pig	NE	1.25 mg/kg i.v. (FPC = 36 ng/mL)	304
	↑iCEB, QRS	2.5 mg/kg i.v. (FPC = 105 ng/mL)	879
	↑ QTcB, Incidence of AV Block (type II/III).	2.5 mg/kg i.v. (FPC = 105 ng/mL)	879

Loperamide’s free TPC (unbound plasma concentration) 0.25 nM drug concentrations in human plasma at steady state C_max_ after 16 mg orally q.d. (Doser et al., 1995 [65]) was used for margin calculations; xFTPC: fold over human FTPC. FPC: free plasma concentration. NE: no relevant effects; APs: cardiac action potentials. ↓↑: significant increase or decrease.

## Data Availability

Raw data in this paper are available upon request.

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
