# Peer review of "The Potential Mechanisms behind Loperamide-Induced Cardiac Arrhythmias Associated with Human Abuse and Extreme Overdose"

_biomolecules, 2023, doi:10.3390/biom13091355_

Round 1

Reviewer 1 Report

The present study is motivated by the increasing reports on cardiac toxicity following intentional ingestion of large overdoses of loperamide, a peripheral opioid receptor agonist normally used to treat diarrhea. The aim of the work was to shed light on the mechanism involved in the acute cardiac electrophysiological toxicity of loperamide at therapeutic and supra-therapeutic doses, by using non-clinical experimental models.

The experimental approach is integrative, combining in vitro patch-clamp electrophysiology, ex vivo model of perfused ventricular wedge, in vivo guinea-pig model and in silico modulization.

The literature on the effects of loperamide on cardiac electrophysiology is abundant, including recent studies using modelisation and similar in vivo models. Still, the present study offers a comprehensive and integrated view on the action of loperamide not only on hERG channel and cardiac repolarization, but also on the inhibition of Na+ and Ca2+ channels and the consequences regarding simulated AP and ECG features. Interestingly, the authors have assessed and discussed the therapeutic margin by comparing their data with data obtained in human database and by estimating experimentally the free plasmatic concentration.

The experiments are well designed and the methodology is relevant. The manuscript is overall clear and sound.

I will only have minor comments on form and discussion.

MATERIAL AND METHODS

Please include the presentation of methods related to general statistics in a specific section.

Also include a “compounds” sections to move the details related to the preparation of loperamide and other compounds (currently in section 2.6).

Regarding experiments involving procedures on animals, please mention a project approval identification if relevant. The legal text sound old (1984); reference directive is 2010/63/EU dating  from 2010.

There are many punctuation signs missing although the manuscript and some spelling mistakes (e.g. legend in Suppl. Table 8). Please scan carefully the whole manuscript and correct.

Some values related to hydroxyzin in Table 1 are centered while others are left aligned .

Some abbreviations should be defined more clearly : I(TA),...

Section 2.3 : “endo-AP” are presented in figure 3D, 3E, but not explained in the methods.

RESULTS

Legend of Figure 1 is not complete; please add N numbers. The text mention n=27, but this is misleading, since there are not 27 replicated for each concentration; please give the n (independent experiments) interval for each doses  e.g: n=3-5 for 1A…

Fig2 : resolution/image quality of the figure is poor. One cannot not read the legend;  The frame or axes do not look upright? If necessary define the legends of the axis in the Figure legend, not in the main text.

P8 : “and produced physiologically relevantly prolongation of AP duration and elicited early afterdepolarization (EAD) at 1200x FTPC.”

Please clarify according the data:

- prolongation of AP duration: from x800

- EAD occurrence: from 1400 (not x1200)

P9, first lane : be more specific: “from x1 to x3 hydoxyzin, respectively”

P9: “The effects of loperamide, alone and in combination with hydroxyzine and metoprolol (a negative control) on cardiac electrophysiology are provided in detail in Supplement table 3-6. » Not clear why this is mentioned here : I can only see some parameters obtained from patch clamp data; used as parameters of the in silico model; this does not look like data presenting the effect of the drug; Please clarify why these tables are presented.

P10: Authors refer to data previously obtained with another model, which are not (yet?) published/ not shown. Because there is not mention of intention for publishing these data elsewhere, this cannot be considered as “unpublished data”, but rather sounds like “data not shown”, which is to be avoided. Please suppress, or rephrase this statement, or add more details on these data (including presentation of the relevant methodology). Please refer to Author Guidelines.

Please define QTcB

P12: Fig 4:  please complete the legend of the figure A, B, … J: explain ECG traces; in general, do not repeat/comment the results in the legend o the figure, this is redundant with main text; at most, mention briefly the main effect (in title of the figure).

DISCUSSION

Regarding data obtained in 3.2 : are there any data from intracellular electrode recording in the literature, that could be compared with these in silico data?

The effect of loperamide on QT interval in the ex vivo rabbit model seem to be at odds with the effect in the in vivo guinea pig model. This is not well discussed. Could that be due to an inhibition of IK(r) by the metabolite in vivo? Or any other effect of the metabolite(s)?

Some toxic effects of chronic exposition to loperamide may be related to change in expression/maturation of ion channels or related proteins. Please comment on this. This may be included as a limitation of the study.

P15: “In addition, drug misusers often took P-glycoprotein inhibitors to increase brain levels of loperamide to achieve an opiate “high” (eg. ketoconazole, fluoxetine, citalopram, omeprazole, quinine, verapamil, erythromycin, Hydroxyzine). Many of these P-glycoprotein inhibitors also directly affect ventricular depolarization and repolarization (QT-interval) (Wu and Juurlink 2017) (Powell and Presnell 2019).”: =>  move to item 3?

The study by Sheng et al., 2017 seems only focused on the effects explained by herg channel inhibition; how could that match with inhibition of AP upstroke in their in silico model ?

The originality of this study over previous studies like Sheng et al, and Takahara et al., should be better emphasized.

PharmaPendium DATABASE

P11: “Abnormal ECG complexes are indicative of conduction disturbances (axis deviation, BBB, P-on-T) were only observed in loperamide-treated animals at 2.5 and 5 mg/kg (free plasma level = 105 and 454 ng/ml, respectively; 879x and 3802x human FTPC, respectively). There is a syntax issue here: “… ECG complexes, WHICH are indicative of (…), were only observed…

Author Response

Q) Please include the presentation of methods related to general statistics in a specific section.

R) Due to the differences between the assays, we have kept the statistical analysis together with their respective models.

Q) Also include a “compounds” sections to move the details related to the preparation of loperamide and other compounds (currently in section 2.6).

R) We have included section 2.1. Compounds.

Q) Regarding experiments involving procedures on animals, please mention a project approval identification if relevant. The legal text sound old (1984); reference directive is 2010/63/EU dating  from 2010.

R) We updated it in the manuscript.

Q) There are many punctuation signs missing although the manuscript and some spelling mistakes (e.g. legend in Suppl. Table 8). Please scan carefully the whole manuscript and correct.

R) Thank you. We went through the manuscript and corrected the punctuation.

Q) Some values related to hydroxyzin in Table 1 are centered while others are left aligned .

R) Indeed, we corrected the table, so the values are all centered.

Q) Some abbreviations should be defined more clearly : I(TA),...

R) This is now detailed.

Q) Section 2.3 : “endo-AP” are presented in figure 3D, 3E, but not explained in the methods.

R) We have included the details in the methods.

RESULTS

Q) Legend of Figure 1 is not complete; please add N numbers. The text mention n=27, but this is misleading, since there are not 27 replicated for each concentration; please give the n (independent experiments) interval for each doses  e.g: n=3-5 for 1A…

R) We have added the n number to the legends.

Q) Fig2 : resolution/image quality of the figure is poor. One cannot not read the legend;  The frame or axes do not look upright? If necessary define the legends of the axis in the Figure legend, not in the main text.

R) We have worked on the figure and updated it in the manuscript. However due to the figures coming from a commercial tool (Oxford in silico tool) we could not change the size of the font inside of all the graphs. 

Q) P8 : “and produced physiologically relevantly prolongation of AP duration and elicited early afterdepolarization (EAD) at 1200x FTPC.”

Please clarify according the data:

- prolongation of AP duration: from x800

- EAD occurrence: from 1400 (not x1200)

P9, first lane : be more specific: “from x1 to x3 hydoxyzin, respectively”

R) We have corrected the manuscript accordingly.

Q) P9: “The effects of loperamide, alone and in combination with hydroxyzine and metoprolol (a negative control) on cardiac electrophysiology are provided in detail in Supplement table 3-6. » Not clear why this is mentioned here: I can only see some parameters obtained from patch clamp data; used as parameters of the in silico model; this does not look like data presenting the effect of the drug; Please clarify why these tables are presented.

R) We have modified the text accordingly.

Q) P10: Authors refer to data previously obtained with another model, which are not (yet?) published/ not shown. Because there is not mention of intention for publishing these data elsewhere, this cannot be considered as “unpublished data”, but rather sounds like “data not shown”, which is to be avoided. Please suppress, or rephrase this statement, or add more details on these data (including presentation of the relevant methodology). Please refer to Author Guidelines.

R) Thank you, We have removed this part.

Q) Please define QTcB

R) This was added to the first mentioned of QTcB P11.

Q) P12: Fig 4:  please complete the legend of the figure A, B, … J: explain ECG traces; in general, do not repeat/comment the results in the legend o the figure, this is redundant with main text; at most, mention briefly the main effect (in title of the figure).

 R) We corrected the legend accordingly.

DISCUSSION

Q) Regarding data obtained in 3.2 : are there any data from intracellular electrode recording in the literature, that could be compared with these in silico data?

R) Thank you for your question. Yes, there are, however, it would be too complex to add in the present manuscript.

Q) The effect of loperamide on QT interval in the ex vivo rabbit model seem to be at odds with the effect in the in vivo guinea pig model. This is not well discussed. Could that be due to an inhibition of IK(r) by the metabolite in vivo? Or any other effect of the metabolite(s)?

R) The effect of loperamide on QT interval in the ex vivo rabbit model seems to be at odds with the effect in the in vivo guinea pig model. In the guinea-pig in vivo, the prolongation of JT-interval (the depolarisation of the heart) was only observed at the top dose and the prolongation of QTcB (both JT-interval+ QRS duration) only at the last two doses. Therefore, the effect of loperamide on QT-interval was co-related well on the effects of QT and JT-interval in the rabbit wedge assay.

Q) Some toxic effects of chronic exposition to loperamide may be related to change in expression/maturation of ion channels or related proteins. Please comment on this. This may be included as a limitation of the study.

R) We haven’t looked at that point and have not found evidence that would suggest the change in expression. However, it is highly possible to have a change in expression of ion channels after long exposure with inhibitors.

Q) P15: “In addition, drug misusers often took P-glycoprotein inhibitors to increase brain levels of loperamide to achieve an opiate “high” (eg. ketoconazole, fluoxetine, citalopram, omeprazole, quinine, verapamil, erythromycin, Hydroxyzine). Many of these P-glycoprotein inhibitors also directly affect ventricular depolarization and repolarization (QT-interval) (Wu and Juurlink 2017) (Powell and Presnell 2019).”: =>  move to item 3?

R) The study by Sheng et al., 2017 seems only focused on the effects explained by herg channel inhibition; how could that match with inhibition of AP upstroke in their in silico model?

In discussion in 4.3. stated as below: Sheng metal 2017: did not use AP upstroke.

The exact mechanisms of cardiac arrhythmias associated with abuse and extreme overdoses of loperamide are unclear but are likely linked to loperamide’s inhibitory effects on cardiac ion-channels at high concentrations. Inhibition of hERG is the most common mechanism for drug-induced QT prolongation and in some situations can lead to the induction of TdP (Sanguinetti, Jiang et al. 1995, Roden 2004). Our preclinical studies have confirmed that loperamide inhibits hERG-mediated IKr in HEK293 cells with an IC50 of 390 nM (based on nominal concentrations). In other published studies, loperamide was reported to inhibit hERG at lower IC50 values from 33-89 nM (Kang, Compton et al. 2016, Klein, Haigney et al. 2016, Sheng, Tran et al. 2017) under different study conditions and measurement protocols.

The originality of this study over previous studies like Sheng et al, and Takahara et al., should be better emphasized.

PharmaPendium DATABASE

Comments on the Quality of English Language

Q) P11: “Abnormal ECG complexes are indicative of conduction disturbances (axis deviation, BBB, P-on-T) were only observed in loperamide-treated animals at 2.5 and 5 mg/kg (free plasma level = 105 and 454 ng/ml, respectively; 879x and 3802x human FTPC, respectively). There is a syntax issue here: “… ECG complexes, WHICH are indicative of (…), were only observed…

R) Corrected accordingly.

Reviewer 2 Report

Loperamide has been used for antidiarrheal treatment for decades but has aroused attention due to increasing side effects, one of which is cardiotoxicity. This study focused on the mechanisms underlying potential heart disorders using in vitro and in vivo experiments. Loperamide was shown in this study to inhibit the cardiac Ikr, Ina and Ica currents, however, the authors seem to miss another cardiac repolarization current IKs. As loperamide affects different kinds of cardiac currents, the mechanism by which it leads to heart disorders can be complicated. Therefore, it is challenging to develop treatments targeting loperamide. In addition, loperamide reduced the QT interval and led to cardiac arrhythmias in isolated rabbit ventricles. Excessive loperamide slows the QRS duration in guinea pigs. A highlight of the manuscript is that it also tested the treatment of loperamide and hydroxyzine together (used in the clinic) on the simulated action potentials in silico modeling. The combination of loperamide and hydroxyzine seems to reduce abnormal action potential intervals, compared to loperamide alone. This study is exciting and would attract a wide range of readers. The manuscript is well-written and organized. However, the blurry figures and small text in them are one of the major concerns, especially in Figure 2. 

None. 

Author Response

Thank you for your review and comments. 

We have worked on the figure 2 and updated it in the manuscript. However, due to the graphs coming out from the commercial in silico tool, we are not able to change the font of the axis within the figure. We added a scale and improve the overall figure.